# Pareto Navigation Gradient Descent: a First-Order Algorithm for Optimization in Pareto Set

**Mao Ye**[1]           **Qiang Liu**[1]

[1]Computer Science Dept., The University of Texas at Austin.

## Abstract

Many modern machine learning applications, such as multi-task learning, require finding optimal model parameters to trade-off multiple objective functions that may conflict with each other. The notion of the Pareto set allows us to focus on the set of (often infinite number of) models that cannot be strictly improved. But it does not provide an actionable procedure for picking one or a few special models to return to practical users. In this paper, we consider *optimization in Pareto set (OPT-in-Pareto)*, the problem of finding Pareto models that optimize an extra reference criterion function within the Pareto set. This function can either encode a specific preference from the users, or represent a generic diversity measure for obtaining a set of diversified Pareto models that are representative of the whole Pareto set. Unfortunately, despite being a highly useful framework, efficient algorithms for OPT-in-Pareto have been largely missing, especially for large-scale, non-convex, and non-linear objectives in deep learning. A naive approach is to apply Riemannian manifold gradient descent on the Pareto set, which yields a high computational cost due to the need for eigen-calculation of Hessian matrices. We propose a first-order algorithm that approximately solves OPT-in-Pareto using only gradient information, with both high practical efficiency and theoretically guaranteed convergence property. Empirically, we demonstrate that our method works efficiently for a variety of challenging multi-task-related problems.

## 1 INTRODUCTION

Although machine learning tasks are traditionally framed as optimizing a single objective, many modern applications, especially in areas like multitask learning, require finding optimal model parameters to minimize multiple objectives (or tasks) simultaneously. As the different objective functions may inevitably conflict with each other, the notion of optimality in multi-objective optimization (MOO) needs to be characterized by the Pareto set: the set of model parameters whose performance of all tasks cannot be jointly improved.

Focusing on the Pareto set allows us to filter out models that can be strictly improved. However, the Pareto set typically contains an infinite number of parameters that represent different trade-offs of the objectives. For $m$ objectives $\ell_1, \ldots, \ell_m$, the Pareto set is often an $(m-1)$ dimensional manifold. It is both intractable and unnecessary to give practical users the whole exact Pareto set. A more practical demand is to find some user-specified special parameters in the Pareto set, which can be framed into the following *optimization in Pareto set (OPT-in-Pareto)* problem:

*Finding one or a set of parameters inside the Pareto set of $\ell_1, \ldots, \ell_m$ that minimize a reference criterion $F$.*

Here the criterion function $F$ can be used to encode an *informative* user-specific preference on the objectives $\ell_1, \ldots, \ell_m$, which allows us to provide the best models customized for different users. $F$ can also be an *non-informative* measure that encourages, for example, the diversity of a set of model parameters. In this case, optimizing $F$ in Pareto set gives a set of diversified Pareto models that are representative of the whole Pareto set, from which different users can pick their favorite models during the testing time.

OPT-in-Pareto provides a highly generic and actionable framework for multi-objective learning and optimization. However, efficient algorithms for solving OPT-in-Pareto have been largely lagging behind in deep learning where the objective functions are non-convex and non-linear. Although has not been formally studied, a straightforward approach is to apply manifold gradient descent on $F$ in the Riemannian manifold formed by the Pareto set [Hillermeier, 2001, Bonnabel, 2013]. However, this casts prohibitive com-

*Accepted for the 38th Conference on Uncertainty in Artificial Intelligence* (UAI 2022).

putational cost due to the need for eigen-computation of Hessian matrices of $\{\ell_i\}$. In the optimization and operation research literature, there has been a body of work on OPT-in-Pareto viewing it as a special bi-level optimization problem [Dempe, 2018]. However, these works often heavily rely on the linearity and convexity assumptions and are not applicable to the non-linear and non-convex problems in deep learning; see for examples in Ecker and Song [1994], Jorge [2005], Thach and Thang [2014], Liu and Ehrgott [2018], Sadeghi and Mohebi [2021] (just to name a few). In comparison, the OPT-in-Pareto problem seems to be much less known and under-explored in the deep learning literature. The exceptions are three works [Mahapatra and Rajan, 2020, Kamani et al., 2021, Chen et al., 2021] that propose specialized algorithms for some specific instantiations of the OPT-in-Pareto problem and we defer a more detailed review to Section 6.

In this work, we provide a practically efficient first-order algorithm for OPT-in-Pareto, using only gradient information of the criterion $F$ and objectives $\{\ell_i\}$. Our method, named *Pareto navigation gradient descent* (PNG), iteratively updates the parameters following a direction that carefully balances the descent on $F$ and $\{\ell_i\}$, such that it guarantees to move towards the Pareto set of $\{\ell_i\}$ when it is far away, and optimize $F$ in a neighborhood of the Pareto set. Our method is simple, practically efficient and has theoretical guarantees.

In empirical studies, we demonstrate that our method works efficiently for both optimizing user-specific criteria and diversity measures. In particular, for finding representative Pareto solutions, we propose an energy distance criterion whose minimizers distribute uniformly on the Pareto set asymptotically [Hardin and Saff, 2004], yielding a principled and efficient Pareto set approximation method that compares favorably with recent works such as Lin et al. [2019], Mahapatra and Rajan [2020]. We also apply PNG to improve the performance of JiGen [Carlucci et al., 2019b], a multi-task learning approach for domain generalization, by using the adversarial feature discrepancy as the criterion objective.

## 2 BACKGROUND ON MULTI-OBJECTIVE OPTIMIZATION

We introduce the background on multi-objective optimization (MOO) and Pareto optimality. For notation, we denote by $[m]$ the integer set $\{1, 2, ...., m\}$, and $\mathbb{R}_+$ the set of non-negative real numbers. Let $\mathcal{C}^m = \left\{\omega \in \mathbb{R}_+^m, \ \sum_{i=1}^m \omega_i = 1\right\}$ be the probability simplex. We denote by $\|\cdot\|$ the Euclidean norm.

Let $\theta \in \mathbb{R}^n$ be a parameter of interest (e.g., the weights in a deep neural network). Let $\boldsymbol{\ell}(\theta) = [\ell_1(\theta), \ldots, \ell_m(\theta)]$ be a set of objective functions that we want to minimize.

For two parameters $\theta, \theta' \in \mathbb{R}^n$, we write $\boldsymbol{\ell}(\theta) \succeq \boldsymbol{\ell}(\theta')$ if $\ell_i(\theta) \geq \ell_i(\theta')$ for all $i \in [m]$; and write $\boldsymbol{\ell}(\theta) \succ \boldsymbol{\ell}(\theta')$ if $\boldsymbol{\ell}(\theta) \succeq \boldsymbol{\ell}(\theta')$ and $\boldsymbol{\ell}(\theta) \neq \boldsymbol{\ell}(\theta')$. We say that $\theta$ is Pareto dominated (or Pareto improved) by $\theta'$ if $\boldsymbol{\ell}(\theta) \succ \boldsymbol{\ell}(\theta')$. We say that $\theta$ is Pareto optimal on a set $\Theta \subseteq \mathbb{R}^n$, denoted as $\theta \in \mathrm{Pareto}(\Theta)$, if there exists no $\theta' \in \Theta$ such that $\boldsymbol{\ell}(\theta) \succ \boldsymbol{\ell}(\theta')$.

The Pareto global optimal set $\mathcal{P}^{**} := \mathrm{Pareto}(\mathbb{R}^n)$ is the set of points (i.e., $\theta$) which are Pareto optimal on the whole domain $\mathbb{R}^n$. The Pareto local optimal set of $\boldsymbol{\ell}$, denoted by $\mathcal{P}^*$, is the set of points which are Pareto optimal on a neighborhood of itself:

$$\mathcal{P}^* := \{\theta \in \mathbb{R}^n : \text{ there exists a neighborhood } \mathcal{N}_\theta \text{ of } \theta,$$
$$\text{such that } \theta \in \mathrm{Pareto}(\mathcal{N}_\theta)\}.$$

The (local or global) Pareto front is the set of objective vectors achieved by the Pareto optimal points, e.g., the local Pareto front is $\mathcal{F}^* = \{\boldsymbol{\ell}(\theta) : \theta \in \mathcal{P}^*\}$. Because finding global Pareto optimum is intractable for non-convex objectives in deep learning, we focus on Pareto local optimal sets in this work; in the rest of the paper, terms like "Pareto set" and "Pareto optimum" refer to Pareto local optimum by default.

**Pareto Stationary Points**   Similar to the case of single-objective optimization, Pareto local optimum implies a notion of Pareto stationarity defined as follows. Assume $\boldsymbol{\ell}$ is differentiable on $\mathbb{R}^n$. A point $\theta$ is called Pareto stationary if there must exists a set of non-negative weights $\omega_1, \ldots, \omega_m$ with $\sum_{i=1}^m \omega_i = 1$, such that $\theta$ is a stationary point of the $\omega$-weighted linear combination of the objectives: $\ell_\omega(\theta) := \sum_{i=1}^m \omega_i \ell_i(\theta)$. Therefore, the set of Pareto stationary points, denoted by $\mathcal{P}$, can be characterized by

$$\mathcal{P} := \{\theta \in \Theta : g(\theta) = 0\} \tag{1}$$
$$g(\theta) := \min_{\omega \in \mathcal{C}^m} \|\sum_{i=1}^m \omega_i \nabla \ell_i(\theta)\|^2,$$

where $g(\theta)$ is the minimum squared gradient norm of $\ell_\omega$ among all $\omega$ in the probability simplex $\mathcal{C}^m$ on $[m]$. Because $g(\theta)$ can be calculated in practice, it provides an essential way to access Pareto local optimality. Being a Pareto stationary point is a necessary condition of being a Pareto local optimum.

**Finding Pareto Optimal Points**   A main focus of the MOO literature is to find a (set of) Pareto optimal points. The simplest approach is *linear scalarization*, which minimizes $\ell_\omega$ for some weight $\omega$ (decided, e.g., by the users) in $\mathcal{C}^m$. However, linear scalarization can only find Pareto points that lie on the *convex envelop* of the Pareto front [see e.g., Boyd et al., 2004], and hence does not give a complete profiling of the Pareto front when the objective functions (and hence their Pareto front) are non-convex.

*Multiple gradient descent (MGD)* [Désidéri, 2012] is an gradient-based algorithm that can converge to a Pareto local optimum that lies on either the convex or non-convex parts of the Pareto front, depending on the initialization. MGD starts from some initialization $\theta_0$ and updates $\theta$ at the $t$-th iteration by

$$\theta_{t+1} \leftarrow \theta_t - \xi v_t, \qquad (2)$$
$$v_t := \arg\max_{v \in \mathbb{R}^n} \left\{ \min_{i \in [m]} \nabla \ell_i(\theta_t)^\top v - \frac{1}{2} \|v\|^2 \right\},$$

where $\xi$ is the step size and $v_t$ is an update direction that maximizes the *worst* descent rate among all objectives, since $\nabla \ell_i(\theta_t)^\top v \approx (\ell_i(\theta_t) - \ell_i(\theta_t - \xi v))/\xi$ approximates the descent rate of objective $\ell_i$ when following direction $v$. When using a sufficiently small step size $\xi$, MGD ensures to yield a *Pareto improvement* (i.e, decreasing all the objectives) on $\theta_t$ unless $\theta_t$ is Pareto (local) optimal; this is because the optimization in Equation (2) always yields $\min_{i \in [m]} \nabla \ell_i(\theta_t)^\top v_t \le 0$ (otherwise we can simply flip the sign of $v_t$).

Using Lagrange strong duality, the solution of Equation (2) can be framed into

$$v_t = \sum_{i=1}^m \omega_{i,t} \nabla \ell_i(\theta_t), \qquad (3)$$
$$\text{where } \{\omega_{i,t}\}_{i=1}^m = \arg \min_{\omega \in \mathcal{C}^m} \|\nabla_\theta \ell_\omega(\theta_t)\|.$$

It is easy to see from Equation (3) that the set of fixed points of MDG (which satisfy $v_t = 0$) coincides with the Pareto stationary set $\mathcal{P}^*$.

A key disadvantage of MGD, however, is that the Pareto point that it converges to depends on the initialization and other algorithm configurations in a rather implicated and complicated way. It is difficult to explicitly control MGD to make it converge to points with specific properties.

## 3  OPTIMIZATION IN PARETO SET

The Pareto set typically contains an infinite number of points. In the *optimization in Pareto set* (OPT-in-Pareto) problem, we are given an extra criterion function $F(\theta)$ in addition to the objectives $\ell$, and we want to minimize $F$ in the Pareto set of $\ell$, that is,

$$\min_{\theta \in \mathcal{P}^*} F(\theta). \qquad (4)$$

For example, one can find the Pareto point whose loss vector $\ell(\theta)$ is the closest to a given reference point $r \in \mathbb{R}^m$ by choosing $F(\theta) = \|\ell(\theta) - r\|^2$. We can also design $F$ to encourages $\ell(\theta)$ to be proportional to $r$, i.e., $\ell(\theta) \propto r$; a constraint variant of this problem was considered in Mahapatra and Rajan [2020].

We can further generalize OPT-in-Pareto to allow the criterion $F$ to depend on an ensemble of Pareto points $\{\theta_1, ..., \theta_N\}$ jointly, that is,

$$\min_{\theta_1, ..., \theta_N \in \mathcal{P}^*} F(\theta_1, ..., \theta_N). \qquad (5)$$

For example, if $F(\theta_1, \ldots, \theta_N)$ measures the diversity among $\{\theta_i\}_{i=1}^N$, then optimizing it provides a set of diversified points inside the Pareto set $\mathcal{P}^*$ yielding a good approximation of $\mathcal{P}^*$. An example of diversity measure is

$$F(\theta_1, \ldots, \theta_N) = E(\ell(\theta_1), \ldots, \ell(\theta_N)), \qquad (6)$$
$$\text{with } E(\ell_1, \ldots, \ell_N) = \sum_{i \ne j} \|\ell_i - \ell_j\|^{-2},$$

where $E$ is known as an *energy distance* in computational geometry, whose minimizer can be shown to give an uniform distribution on manifold asymptotically when $N \to \infty$ [Hardin and Saff, 2004]. This formulation is particularly useful when the users' preference is unknown during the training time, and we want to return an ensemble of models that well cover the different areas of the Pareto set to allow the users to pick up a model that fits their needs regardless of their preference. The problem of profiling Pareto set has attracted a line of recent works [e.g., Lin et al., 2019, Mahapatra and Rajan, 2020, Ma et al., 2020, Deist et al., 2021], but they rely on specific criterion or heuristics and do not address the general optimization of form Equation (5).

**Manifold Gradient Descent**  One straightforward approach to OPT-in-Pareto is to deploy manifold gradient descent [Hillermeier, 2001, Bonnabel, 2013], which conducts steepest descent of $F(\theta)$ in the Riemannian manifold formed by the Pareto set $\mathcal{P}^*$. Initialized at $\theta_0 \in \mathcal{P}^*$, manifold gradient descent updates $\theta_t$ at the $t$-th iteration along the direction of the projection of $\nabla F(\theta_t)$ on the tangent space $\mathcal{T}(\theta_t)$ at $\theta_t$ in $\mathcal{P}^*$,

$$\theta_{t+1} = \theta_t - \xi \text{Proj}_{\mathcal{T}(\theta_t)}(\nabla F(\theta_t)).$$

By using the stationarity characterization in Equation (1), under proper regularity conditions, one can show that the tangent space $\mathcal{T}(\theta_t)$ equals the null space of the Hessian matrix $\nabla_\theta^2 \ell_{\omega_t}(\theta_t)$, where $\omega_t = \arg\min_{\omega \in \mathcal{C}^m} \|\nabla_\theta \ell_\omega(\theta_t)\|$. However, the key issue of manifold gradient descent is the high cost for calculating this null space of Hessian matrix. Although numerical techniques such as Krylov subspace iteration [Ma et al., 2020] or conjugate gradient descent [Koh and Liang, 2017] can be applied, the high computational cost (and the complicated implementation) still impedes its application in large scale deep learning problems. See Section 1 for discussions on other related works.

## 4 PARETO NAVIGATION GRADIENT DESCENT FOR OPT-IN-PARETO

We now introduce our main algorithm, Pareto Navigating Gradient Descent (PNG), which provides a practical approach to OPT-in-Pareto. For convenience, we focus on the single point problem in Equation (4) in the presentation. The generalization to the multi-point problem in Equation (5) is straightforward. We first introduce the main idea and then present theoretical analysis in Section 5.

We consider the general incremental updating rule of form

$$\theta_{t+1} \leftarrow \theta_t - \xi v_t,$$

where $\xi$ is the step size and $v_t$ is an update direction that we shall choose to achieve the following desiderata in balancing the decent of $\{\ell_i\}$ and $F$:

i) When $\theta_t$ is far away from the Pareto set, we want to choose $v_t$ to give Pareto improvement to $\theta_t$, moving it towards the Pareto set. The amount of Pareto improvement might depend on how far $\theta_t$ is to the Pareto set.

ii) If the directions that yield Pareto improvement are not unique, we want to choose the Pareto improvement direction that decreases $F(\theta)$ most.

iii) When $\theta_t$ is very close to the Pareto set, e.g., having a small $g(\theta)$, we want to fully optimize $F(\theta)$.

We achieve the desiderata above by using the $v_t$ that solves the following optimization:

$$v_t = \underset{v \in \mathbb{R}^n}{\arg\min} \left\{ \frac{1}{2} \|\nabla F(\theta_t) - v\|^2 \right\} \tag{7}$$
$$\text{s.t. } \nabla_\theta \ell_i(\theta_t)^\top v \geq \phi_t, \quad \forall i \in [m],$$

where we want $v_t$ to be as close to $\nabla F(\theta_t)$ as possible (hence decrease $F$ most), conditional on that the decreasing rate $\nabla_\theta \ell_i(\theta_t)^\top v_t$ of all losses $\ell_i$ are lower bounded by a *control parameter* $\phi_t$. A positive $\phi_t$ enforces that $\nabla_{\theta_t} \ell_i(\theta)^\top v_t$ is positive for all $\ell_i$, hence ensuring a Pareto improvement when the step size is sufficiently small. The magnitude of $\phi_t$ controls how much Pareto improvement we want to enforce, so we may want to gradually decrease $\phi_t$ when we move closer to the Pareto set. In fact, varying $\phi_t$ provides an intermediate updating direction between the vanilla gradient descent on $F$ and MGD on $\{\ell_i\}$:

i) If $\phi_t = -\infty$, we have $v_t = \nabla F(\theta_t)$ and it conducts a pure gradient descent on $F$ without considering $\{\ell_i\}$.

ii) If $\phi_t \to +\infty$, then $v_t$ approaches to the MGD direction of $\{\ell_i\}$ in Equation (2) without considering $F$.

In this work, we propose to choose $\phi_t$ based on the minimum gradient norm $g(\theta_t)$ in Equation (1) as a surrogate indication of Pareto local optimality. In particular, we consider the

following simple design:

$$\phi_t = \begin{cases} -\infty & \text{if } g(\theta_t) \leq e, \\ \alpha_t g(\theta_t) & \text{if } g(\theta_t) > e, \end{cases} \tag{8}$$

where $e$ is a small tolerance parameter and $\alpha_t$ is a positive hyper-parameter. When $g(\theta_t) > e$, we set $\phi_t$ to be proportional to $g(\theta_t)$, to ensure Pareto improvement based on how far $\theta_t$ is to Pareto set. When $g(\theta_t) \leq e$, we set $\phi_t = -\infty$ which "turns off" the control and hence fully optimizes $F(\theta)$.

In practice, the optimization in Equation (7) can be solved efficiently by its dual form as follows.

**Theorem 1.** *The solution $v_t$ of Equation (7), if it exists, has a form of*

$$v_t = \nabla F(\theta_t) + \sum_{t=1}^{m} \lambda_{i,t} \nabla \ell_i(\theta_t), \tag{10}$$

*with $\{\lambda_{i,t}\}_{t=1}^{m}$ the solution of the following dual problem*

$$\max_{\lambda \in \mathbb{R}_+^m} -\frac{1}{2} \|\nabla F(\theta_t) + \sum_{i=1}^{m} \lambda_t \nabla \ell_i(\theta_t)\|^2 + \sum_{i=1}^{m} \lambda_i \phi_t. \tag{11}$$

The optimization in Equation (11) can be solved efficiently for a small $m$ (e..g, $m \leq 10$), which is the case for typical applications. We include the details of the practical implementation in Algorithm 1.

## 5 THEORETICAL PROPERTIES

We provide a theoretical quantification on how PNG guarantees to i) move the solution towards the Pareto set (Theorem 2); and ii) optimize $F$ in a neighborhood of Pareto set (Theorem 4). To simplify the result and highlight the intuition, we focus on the continuous time limit of PNG, which yields a differentiation equation $\mathrm{d}\theta_t = -v_t \mathrm{d}t$ with $v_t$ defined in Equation (7), where $t \in \mathbb{R}_+$ is a continuous integration time.

**Assumption 1.** *Let $\{\theta_t \colon t \in \mathbb{R}_+\}$ be a solution of $\mathrm{d}\theta_t = -v_t \mathrm{d}t$ with $v_t$ in Equation (7); $\phi_k$ in Equation (8); $e > 0$; and $\alpha_t \geq 0, \forall t \in \mathbb{R}_+$. Assume $F$ and $\ell$ are continuously differentiable on $\mathbb{R}^n$, and lower bounded with $F^* := \inf_{\theta \in \mathbb{R}^n} F(\theta) > -\infty$ and $\ell_i^* := \inf_{\theta \in \mathbb{R}^n} \ell_i(\theta) > -\infty$. Assume $\sup_{\theta \in \mathbb{R}^n} \|\nabla F(\theta)\| \leq c$.*

Technically, $\mathrm{d}\theta_t = -v_t \mathrm{d}t$ is a piecewise smooth dynamical system whose solution should be taken in the Filippov sense using the notion of differential inclusion [Bernardo et al., 2008]. The solution always exists under mild regularity conditions although it may not be unique. Our results below apply to all solutions.

**Algorithm 1** Pareto Navigating Gradient Descent

---
1: Initialize $\theta_0$; decide the step size $\xi$, and the control function $\phi$ in Equation (8) (including the threshold $e > 0$ and the descending rate $\{\alpha_t\}$).
2: **for** iteration $t$ **do**

$$\theta_{t+1} \leftarrow \theta_t - \xi v_t, \qquad\qquad v_t = \nabla F(\theta_t) + \sum_{i=1}^{m} \lambda_{i,t} \nabla \ell_i(\theta_t), \qquad\qquad (9)$$

where $\lambda_{i,t} = 0$, $\forall i \in [m]$ if $g(\theta_t) \leq e$, and $\{\lambda_{i,t}\}_{t=1}^{m}$ is the solution of (11) with $\phi(\theta_t) = \alpha_t g(\theta_t)$ when $g(\theta_t) > e$.
3: **end for**

---

## 5.1 PARETO OPTIMIZATION

We now show that the algorithm converges to the vicinity of Pareto set quantified by a notion of Pareto closure. For $\epsilon \geq 0$, let $\mathcal{P}_\epsilon$ be the set of Pareto $\epsilon$-stationary points: $\mathcal{P}_\epsilon = \{\theta \in \mathbb{R}^n \colon g(\theta) \leq \epsilon\}$. The Pareto closure of a set $\mathcal{P}_\epsilon$, denoted by $\overline{\mathcal{P}_\epsilon}$ is the set of points that perform no worse than at least one point in $\mathcal{P}_\epsilon$, that is,

$$\overline{\mathcal{P}_\epsilon} := \cup_{\theta \in \mathcal{P}_\epsilon} \overline{\{\theta\}}, \quad \overline{\{\theta\}} = \{\theta' \in \mathbb{R}^n \colon \boldsymbol{\ell}(\theta') \preceq \boldsymbol{\ell}(\theta)\}.$$

Therefore, $\overline{\mathcal{P}_\epsilon}$ is better than or at least as good as $\mathcal{P}_\epsilon$ in terms of Pareto efficiency.

**Theorem 2** (Pareto Improvement on $\boldsymbol{\ell}$). *Under Assumption 1, assume $\theta_0 \notin \mathcal{P}_e$, and $t_e$ is the first time when $\theta_{t_e} \in \mathcal{P}_e$, then for any time $t < t_e$,*

$$\frac{d}{dt}\ell_i(\theta_t) \leq -\alpha_t g(\theta_t), \quad \min_{s \in [0,t]} g(\theta_s) \leq \frac{\min_{i \in [m]}(\ell_i(\theta_0) - \ell_i^*)}{\int_0^t \alpha_s ds}.$$

*Therefore, the update yields Pareto improvement on $\boldsymbol{\ell}$ when $\theta_t \notin \mathcal{P}_e$ and $\alpha_t g(\theta_t) > 0$.*

*Further, if $\int_0^t \alpha_s ds = +\infty$, then for any $\epsilon > e$, there exists a finite time $t_\epsilon \in \mathbb{R}_+$ on which the solution enters $\mathcal{P}_\epsilon$ and stays within $\overline{\mathcal{P}_\epsilon}$ afterwards, that is, we have $\theta_{t_\epsilon} \in \mathcal{P}_\epsilon$ and $\theta_t \in \overline{\mathcal{P}_\epsilon}$ for any $t \geq t_\epsilon$.*

Here we guarantee that $\theta_t$ must enter $\mathcal{P}_\epsilon$ for some time (in fact infinitely often), but it is not confined in $\mathcal{P}_\epsilon$. On the other hand, $\theta_t$ does not leave $\overline{\mathcal{P}_\epsilon}$ after it first enters $\mathcal{P}_\epsilon$ thanks to the Pareto improvement property.

## 5.2 CRITERION OPTIMIZATION

We now show that PNG finds a local optimum of $F$ inside the Pareto closure $\overline{\mathcal{P}_\epsilon}$ in an approximate sense. We first show that a fixed point $\theta$ of the algorithm that is locally convex on $F$ and $\boldsymbol{\ell}$ must be a local optimum of $F$ in the Pareto closure of $\{\theta\}$, and then quantify the convergence of the algorithm.

**Theorem 3** (PNG Finds Local Optimum). *Under Assumption 1, we have*

*If $\theta_t \notin \mathcal{P}_e$ is a fixed point of the algorithm, that is, $\frac{d\theta_t}{dt} = -v_t = 0$, and $F, \boldsymbol{\ell}$ are convex in a neighborhood $\theta_t$, then $\theta_t$ is a local minimum of $F$ in the Pareto closure $\overline{\{\theta_t\}}$, that*

*is, there exists a neighborhood of $\theta_t$ in which there exists no point $\theta'$ such that $F(\theta') < F(\theta_t)$ and $\boldsymbol{\ell}(\theta') \preceq \boldsymbol{\ell}(\theta_t)$.*

*If $\theta_t \in \mathcal{P}_e$, we have $v_t = \nabla F(\theta_t)$, and hence a fixed point with $\frac{d\theta_t}{dt} = -v_t = 0$ is an unconstrained local minimum of $F$ when $F$ is locally convex on $\theta_t$.*

**Theorem 4** (Convergence). *Let $\epsilon > e$ and assume $g_\epsilon := \sup_\theta\{g(\theta)\colon \theta \in \overline{\mathcal{P}_\epsilon}\} < +\infty$ and $\sup_{t \geq 0} \alpha_t < \infty$. Under Assumption 1, when we initialize from $\theta_0 \in \mathcal{P}_\epsilon$, we have*

$$\min_{s \in [0,t]}\left\|\frac{d\theta_s}{ds}\right\|^2 \leq \frac{F(\theta_0) - F^*}{t} + \frac{1}{t}\int_0^t \alpha_s\left(\alpha_s g_\epsilon + c\sqrt{g_\epsilon}\right)ds.$$

*In particular, if we have $\alpha_t = \alpha = const$, then $\min_{s \in [0,t]}\|d\theta_s/ds\|^2 = \mathcal{O}\left(1/t + \alpha\sqrt{g_\epsilon}\right)$.*

*If $\int_0^\infty \alpha_t^\gamma dt < +\infty$ for some $\gamma \geq 1$, we have $\min_{s \in [0,t]}\|d\theta_s/ds\|^2 = \mathcal{O}(1/t + \sqrt{g_\epsilon}/t^{1/\gamma})$.*

Combining the results in Theorem 2 and 4, we can see that the choice of sequence $\{\alpha_t \colon t \in \mathbb{R}_+\}$ controls how fast we want to decrease $\boldsymbol{\ell}$ vs. $F$. Large $\alpha_t$ yields faster descent on $\boldsymbol{\ell}$, but slower descent on $F$. Theoretically, using a sequence that satisfies $\int \alpha_t dt = +\infty$ and $\int \alpha_t^\gamma dt < +\infty$ for some $\gamma > 1$ allows us to ensure that both $\min_{s \in [0,t]} g(\theta_s)$ and $\min_{s \in [0,t]}\|d\theta/ds\|^2$ converge to zero. If we use a constant sequence $\alpha_t = \alpha$, it introduces an $\mathcal{O}(\alpha\sqrt{g_\epsilon})$ term that does not vanish as $t \to +\infty$. However, we can expect that $g_\epsilon$ is small when $\epsilon$ is small for well-behaved functions. In practice, we find that constant $\alpha_t$ works sufficiently well.

# 6 RELATED WORK

**Optimization Algorithms for MOO** There has been a rising interest in MOO in deep learning, mostly in the context of multi-task learning. But most existing methods can not be applied to the general OPT-in-Pareto problem. A large body of recent works focus on improving non-convex optimization for finding *some* model in the Pareto set, but cannot search for a *special* model satisfying a specific criterion [Chen et al., 2018, Kendall et al., 2018, Sener and Koltun, 2018, Yu et al., 2020, Chen et al., 2020, Wu et al., 2020, Fifty et al., 2020, Javaloy and Valera, 2021].

**Specific Instantiations of OPT-in-Pareto** One previous work [Mahapatra and Rajan, 2020] and two concurrent

works [Kamani et al., 2021, Chen et al., 2021] study specific instantiations of the general OPT-in-Pareto problem and thus are highly related to this paper. We give a detailed review. Mahapatra and Rajan [2020] aims to search Pareto model that satisfies a constraint on the ratio between the different objectives, which can be viewed as OPT-in-Pareto problem when the criterion $F$ is a proper measure of constraint violation (i.e, the non-uniformity score defined in Mahapatra and Rajan [2020]). EPO, the proposed algorithm in Mahapatra and Rajan [2020] heavily relies on a special property of the ratio constraint problem: there always exists an updating direction that either gives Pareto improvement or reduces the constraint violation or both. However, a general OPT-in-Pareto problem does not have such nice property, making EPO only a specialized algorithm for the ratio constraint problem rather than a general OPT-in-Pareto problem. In section 7.1 we demonstrate that PNG is able to recover the functionality of EPO while being a more general algorithm for OPT-in-Pareto. [Kamani et al., 2021] formulate the fairness learning as a MOO problem in which the accuracy and fairness measure are considered as the two objectives. It first proposes PDO, an algorithm that converges to Pareto stationary set by viewing MOO as a bi-level optimization (which is a standard MOO algorithm that does not solve any instance of OPT-in-Pareto) and then BP-PDO, an modification of PDO that seeks a Pareto model that satisfies the ratio-constraint considered in Mahapatra and Rajan [2020]. Admittedly, it is possible to extend the BP-PDO for general OPT-in-Pareto problems but such extension is non-trivial: even for the special ratio-constraint problem, it is unclear what convergence and optimality guarantee BP-PDO has (only guarantee of PDO is given in Kamani et al. [2021]). In comparison, our PNG is shown to converge to the local optimum of OPT-in-Pareto problem. Chen et al. [2021] aims to pre-train a multi-task model such that the representations of the tasks are similar. Their problem is essentially an OPT-in-Pareto problem where the discrepancy of task representations are chosen as the criterion function. Compared with PNG, the proposed TAWT algorithm requires the computation of inverse Hessian product at each iteration making its computational cost large.

**Approximation of Pareto Set** There has been increasing interest in finding a compact approximation of the Pareto set. Navon et al. [2020], Lin et al. [2020] use hypernetworks to approximate the map from linear scalarization weights to the corresponding Pareto solutions; these methods could not fully profile non-convex Pareto fronts due to the limitation of linear scalarization [Boyd et al., 2004], and the use of hypernetwork introduces extra optimization difficulty. Another line of works [Lin et al., 2019, Mahapatra and Rajan, 2020] approximate the Pareto set by Pareto models with different user preference vectors that rank the relative importance of different tasks; these methods need a good heuristic design of preference vectors, which requires prior knowledge of the

Pareto front. Ma et al. [2020] leverages manifold gradient to conduct a local random walk on the Pareto set but suffers from the high computational cost. Deist et al. [2021] approximates the Pareto set by maximizing hypervolume, which also requires prior knowledge for a careful choice of good reference vector. Liu et al. [2021] introduces a repulsive force to encourage the model diversity without hurting their Pareto Optimality.

**Applications of MOO** Multi-task learning can also be applied to improve the learning in many other domains including domain generalization [Dou et al., 2019, Carlucci et al., 2019a, Albuquerque et al., 2020], domain adaption [Sun et al., 2019, Luo et al., 2021], model uncertainty [Hendrycks et al., 2019, Zhang et al., 2020, Xie et al., 2021], adversarial robustness [Yang and Vondrick, 2020] and semi-supervised learning [Sohn et al., 2020]. All of those applications utilize a linear scalarization to combine the multiple objectives and it is thus interesting to apply the proposed OPT-in-Pareto framework, which we leave for future work.

# 7 EMPIRICAL RESULTS

We introduce three applications of OPT-in-Pareto with PNG: Singleton Preference, Pareto approximation and improving multi-task based domain generalization method. We also conduct additional study on how the learning dynamics of PNG changes with different hyper-parameters ($\alpha_t$ and $e$), which are included in Appendix **??**. Other additional results that are related to the experiments in Section 7.1 and 7.2 and are included in the Appendix will be introduced later in their corresponding sections. Code is available at https://github.com/lushleaf/ParetoNaviGrad.

## 7.1 FINDING PREFERRED PARETO MODELS

We consider the synthetic example used in Lin et al. [2019], Mahapatra and Rajan [2020], which consists of two losses: $\ell_1(\theta) = 1 - \exp(-\|\theta - \eta\|^2)$ and $\ell_2(\theta) = 1 - \exp(-\|\theta + \eta\|^2)$, where $\eta = n^{-1/2}$ and $n = 10$ is dimension of the parameter $\theta$.

**Ratio-based Criterion** We first show that PNG can solve the search problem under the ratio constraint of objectives in Mahapatra and Rajan [2020], i.e., finding a point $\theta \in \mathcal{P}^* \cap \Omega$ with $\Omega = \{\theta : r_1\ell_1(\theta) = r_2\ell_2(\theta) = ... = r_m\ell_m(\theta)\}$, given some preference vector $r = [r_1, ..., r_m]$. We apply PNG with the non-uniformity score defined in Mahapatra and Rajan [2020] as the criterion, and compare with their algorithm called exact Pareto optimization (EPO). We show in Figure 1(a)-(b) the trajectory of PNG and EPO for searching models with different preference vector $r$, starting from the same randomly initialized point. Both PNG and EPO converge to the correct solutions but with different trajectories. This

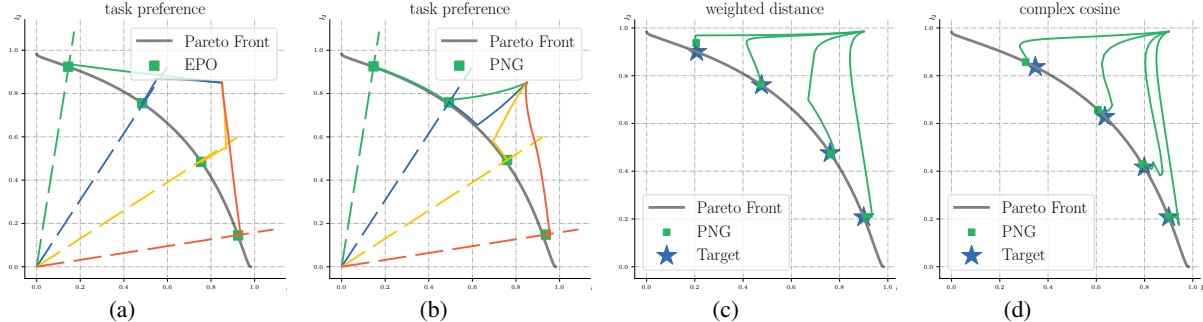

(a)          (b)          (c)          (d)

Figure 1: (a)-(b): the trajectory of finding Pareto models that satisfy different ratio constraints (shown in different colors) on the two objectives $\ell_1, \ell_2$ using EPO and PNG; we can see that PNG can achieve the same goal as EPO (with different trajectories) while being a more general approach. (c)-(d): the trajectory of finding Pareto models that minimize the weighted distance and complex cosine criterion using PNG. The green dots indicate the converged models. We can see that PNG can successfully locate the correct Pareto models that minimize different criteria.

suggests that PNG is able to achieve the same functionality of finding ratio-constraint Pareto models as Mahapatra and Rajan [2020], Kamani et al. [2021] do but being versatile to handle general criteria. We refer readers to Appendix **??** for more results with different choices of hyper-parameters and the experiment details.

**Other Criteria**   We demonstrate that PNG is able to find solutions for general choices of $F$. We consider the following designs of $F$: 1) weighted $\ell_2$ distance w.r.t. a reference vector $r \in \mathbb{R}_+^m$, that is, $F_{\mathrm{wd}}(\theta) = \sum_{i=1}^{m} (\ell_i(\theta) - r_i)^2 / r_i$; and 2) complex cosine: in which $F$ is a complicated function related to the cosine of task objectives, i.e., $F_{\mathrm{cs}} = -\cos\left(\pi(\ell_1(\theta) - r_1)/2\right) + (\cos(\pi(\ell(\theta_2) - r_2)) + 1)^2$. Here the weighted $\ell_2$ distance can be viewed as finding a Pareto model that has the losses close to some target value $r$, which can be viewed as an alternative approach to partition the Pareto set. The design of complex cosine aims to test whether PNG is able to handle a very non-linear criterion function. In both cases, we take $r_1 = [0.2, 0.4, 0.6, 0.8]$ and $r_2 = 1 - r_1$. We show in Fig 1(c)-(d) the trajectory of PNG. As we can see, PNG is able to correctly find the optimal solutions of OPT-in-Pareto. We also test PNG on a more challenging ZDT2-variant used in Ma et al. [2020] and a larger scale MTL problem [Liu et al., 2019], for which we refer readers to Appendix **??** and **??**.

### 7.2 FINDING DIVERSE PARETO MODELS

**Synthetic Examples**   We reuse the synthetic example introduced in Section 7.1. We consider learning 5 models to approximate the Pareto front staring from two types of extremely bad initializations. Specifically, in the upper row of Figure 2, we consider initializing the models using linear scalarization. Due to the concavity of the Pareto front, linear scalarization can only learns models at the two extreme end of the Pareto front. The second row uses MGD for initialization and the models is scattered at an small region of the Pareto front. Different from the algorithm proposed by Lin

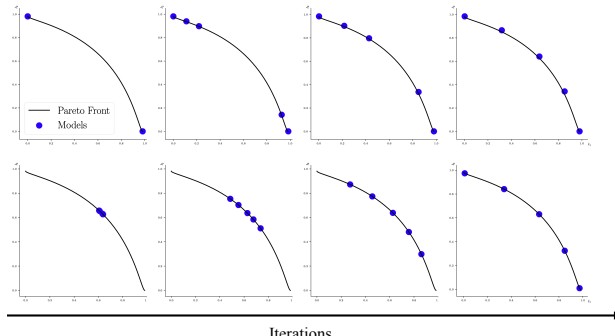

Iterations

Figure 2: Evolution of models from different initializations. Upper row starts with models at the boundary of the Pareto set. Lower row considers clustered initializations.

et al. [2019] which relies on a good initialization, using the proposed energy distance function, PNG pushes the models to be equally distributed on the Pareto Front without the need of any prior information of the Pareto front even with extremely bad starting point.

**Multi-MNIST Benchmark**   We consider the problem of finding diversified points from the Pareto set by minimizing the energy distance criterion in Equation (6). We use the same setting as Lin et al. [2019], Mahapatra and Rajan [2020]. We consider three benchmark datasets: (1) MultiMNIST, (2) MultiFashion, and (3) MultiFashion+MNIST. For each dataset, there are two tasks (classifying the top-left and bottom-right images). We consider LeNet with multihead and train $N = 5$ models to approximate the Pareto set. For baselines, we compare with linear scalarization, MGD [Sener and Koltun, 2018], and EPO [Mahapatra and Rajan, 2020]. For the MGD baseline, we find that naively running it leads to poor performance as the learned models are not diversified and thus we initialize the MGD with 60-epoch runs of linear scalarization with equally distributed preference weights and runs MGD for the later 40 epoch. We refer the reader to Appendix **??** for more details of the

| Data | Method | Loss | | Acc | |
|------|--------|------|------|------|------|
| | | HV↑ ($10^{-2}$) | IGD+↓ ($10^{-2}$) | HV↑ ($10^{-2}$) | IGD+↓ ($10^{-2}$) |
| Multi-MNIST | Linear | $7.48 \pm 0.11$ | $0.142 \pm 0.034$ | $9.27 \pm 0.024$ | $0.036 \pm 0.0084$ |
| | MGD | $7.69 \pm 0.10$ | $0.051 \pm 0.011$ | $9.27 \pm 0.023$ | $0.008 \pm 0.0010$ |
| | EPO | $\mathbf{7.87 \pm 0.16}$ | $0.069 \pm 0.028$ | $9.17 \pm 0.032$ | $0.065 \pm 0.0181$ |
| | PNG | $\mathbf{7.86 \pm 0.11}$ | $\mathbf{0.042 \pm 0.012}$ | $\mathbf{9.39 \pm 0.036}$ | $\mathbf{0.006 \pm 0.0022}$ |
| Multi-Fashion | Linear | $0.38 \pm 0.059$ | $0.127 \pm 0.013$ | $4.76 \pm 0.019$ | $0.064 \pm 0.012$ |
| | MGD | $0.42 \pm 0.064$ | $0.046 \pm 0.016$ | $4.77 \pm 0.019$ | $\mathbf{0.023 \pm 0.003}$ |
| | EPO | $0.36 \pm 0.058$ | $0.308 \pm 0.109$ | $4.78 \pm 0.030$ | $0.211 \pm 0.020$ |
| | PNG | $\mathbf{0.47 \pm 0.066}$ | $\mathbf{0.016 \pm 0.002}$ | $\mathbf{4.81 \pm 0.021}$ | $\mathbf{0.023 \pm 0.003}$ |
| Fashion-MNIST | Linear | $5.01 \pm 0.057$ | $0.167 \pm 0.054$ | $8.46 \pm 0.046$ | $0.110 \pm 0.035$ |
| | MGD | $5.09 \pm 0.069$ | $0.060 \pm 0.029$ | $8.40 \pm 0.045$ | $\mathbf{0.049 \pm 0.011}$ |
| | EPO | $4.60 \pm 0.166$ | $0.233 \pm 0.054$ | $8.12 \pm 0.041$ | $0.385 \pm 0.077$ |
| | PNG | $\mathbf{5.27 \pm 0.054}$ | $0.048 \pm 0.027$ | $\mathbf{8.53 \pm 0.047}$ | $0.046 \pm 0.022$ |

Table 1: Results of approximating the Pareto set by different methods on three MNIST benchmark datasets. The numbers in the table are the averaged value and the standard deviation. Bolded values indicate the statistically significant best result with p-value less than 0.5 based on matched pair t-test.

| PACS | art paint | cartoon | sketches | photo | Avg |
|------|-----------|---------|----------|-------|-----|
| D-SAM | 0.7733 | 0.7243 | 0.7783 | 0.9530 | 0.8072 |
| DeepAll | 0.7785 | 0.7486 | 0.6774 | 0.9573 | 0.7905 |
| JiGen | $0.8009 \pm 0.004$ | $0.7363 \pm 0.007$ | $0.7046 \pm 0.013$ | $\mathbf{0.9629 \pm 0.002}$ | $0.8012 \pm 0.002$ |
| JiGen+adv | $0.7923 \pm 0.006$ | $0.7402 \pm 0.004$ | $0.7188 \pm 0.005$ | $0.9617 \pm 0.001$ | $0.8033 \pm 0.001$ |
| JiGen+PNG | $\mathbf{0.8014 \pm 0.005}$ | $\mathbf{0.7538 \pm 0.001}$ | $\mathbf{0.7222 \pm 0.006}$ | $0.9627 \pm 0.002$ | $\mathbf{0.8100 \pm 0.005}$ |

Table 2: Comparing different methods for domain generalization on PACS using ResNet-18. The values in table are the testing accuracy with its standard deviation. The bolded values are the best models with p-value less than 0.1 based on match-pair t-test.

experiments.

We measure the quality of how well the found models $\{\theta_1, \ldots, \theta_N\}$ approximate the Pareto set using two standard metrics: Inverted Generational Distance Plus (IGD+) [Ishibuchi et al., 2015] and hypervolume (HV) [Zitzler and Thiele, 1999]; see Appendix **??** for their definitions. We run all the methods with 5 independent trials and report the averaged value and its standard deviation in Table 1. We report the scores calculated based on loss (cross-entropy) and accuracy on the test set. The bolded values indicate the best result with p-value less than 0.05 (using matched pair t-test). In most cases, PNG improves the baselines by a large margin. We include ablation studies in Appendix **??** and additional comparisons with the second-order approach proposed by Ma et al. [2020] in Appendix **??**.

## 7.3 APPLICATION TO MULTI-TASK BASED DOMAIN GENERALIZATION ALGORITHM

JiGen [Carlucci et al., 2019b] learns a domain generalizable model by learning two tasks based on linear scalarization, which essentially searches for a model in the Pareto set and requires choosing the weight of linear scalarization carefully. It is thus natural to study whether there is a better

mechanism that dynamically adjusts the weights of the two losses so that we eventually learn a better model. Motivated by the adversarial feature learning [Ganin et al., 2016], we propose to improve JiGen such that the latent feature representations of the two tasks are well aligned. This can be framed into an OPT-in-Pareto problem where the criterion is the discrepancy of the latent representations (implemented using an adversarial discrepancy module in the network) of the two tasks. PNG is applied to solve the optimization. We evaluate the methods on PACS [Li et al., 2017], which covers 7 object categories and 4 domains (Photo, Art Paintings, Cartoon, and Sketches). The model is trained on three domains and tested on the rest of them. Our approach is denoted as JiGen+PNG and we also include JiGen + adv, which simply adds the adversarial loss as regularization and two other baseline methods (D-SAM [D'Innocente and Caputo, 2018] and DeepAll [Carlucci et al., 2019b]). For the three JiGen based approaches, we run 3 independent trials and for the other two baselines, we report the results in their original papers. Table 2 shows the result using ResNet-18, which demonstrates the improvement by the application of the OPT-in-Pareto framework. We also include the results using AlexNet in the Appendix. Please see Appendix **??** for the additional results and more experiment details.

# 8  CONCLUSION

This paper studies the OPT-in-Pareto, a problem that has been studied in operation research with restrictive linear or convexity assumption but largely under-explored in deep learning literature, in which the objectives are non-linear and non-convex. Applying algorithms such as manifold gradient descent requires eigen-computation of the Hessian matrix at each iteration and thus can be expensive. We propose a first-order approximation algorithm called Pareto Navigation Gradient Descent (PNG) with theoretically guaranteed descent and convergence property to solve OPT-in-Pareto.

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
