# OpenReview forum: "Pareto Navigation Gradient Descent: a First-Order Algorithm for Optimization in Pareto Set"
_auai.org/UAI/2022/Conference — UAI 2022 Poster_

### Official Review · Reviewer_951o · 2022-04-01

**Q2(1) Originality/Novelty:** 3
**Q2(2) Significance/Impact:** 2
**Q2(3) Correctness/Technical Quality:** 3
**Q2(6) Clarity Of Writing:** 3
**Q6 Overall Score:** 7
**Q8 Confidence In Your Score:** 2

**Q1 Summary And Contributions:**

The paper proposes an approach to optimize a reference criterion within the set of Pareto-optimal points.
It provides a first-order algorithm that can solve this problem given the gradient information of the reference criterion and objectives.
The paper then provides proofs for properties of the proposed algorithm and demonstrates its efficacy in a benchmark study.


**Q2 Assessment Of The Paper:**

More detailed information regarding each of these aspects is given below:

**Q2(4) Quality Of Experiments (Optional):**

3: Good: The experimental evaluation is adequate, and the results convincingly support the main claims.

**Q2(5) Reproducibility:**

2: Fair: Key resources (e.g., proofs, code, data) are unavailable but key details (e.g., proof sketches, experimental setup) are sufficiently well-described for an expert to confidently reproduce the main results.

**Q3 Main Strengths:**

The paper tackles an important problem: Allowing for a diverse set of user preferences in multi-objective optimization procedures.

The paper is generally well written and clearly structured. It introduces the problem and relevant other approaches, proposes the solution and provides a few proofs of relevant properties.

The technical contribution is - as far as I can judge - novel and provide a relevant improvement over existing work

The supplementary is rich and provides proofs as well as relevant experimental details.

**Q4 Main Weakness:**

The paper does not provide any code for reproducibility. Results seem surprisingly good, so code would be highly appreciated.

The paper introduces several hyperparameters alpha and threshold epsilon.
I am missing a discussion on the effect and choice of those hyperparameters.

The paper has quite a few typos and grammatical mistakes, e.g. "if there must exists a set of negative weights".
This sometimes makes the paper harder to understand than needed.

**Q5 Detailed Comments To The Authors:**

The title is odd, in the sense that it is unclear what "special" is in this context. Consider perhaps adapting the title.


The exposition could be slightly adapted to improve clarity:
  The introduction assumes that the reader is somewhat familiar with the relevant literature and could be made more accessible by
  e.g. providing a small example

  The experiments (multiMNIST) do not state clearly what the goal of the algorithm is, e.g. with respect to which criteria improvement in HV is judged.

The paper proposes a method that only relies on gradient information, stating that previous methods (here Manifold Gradient Descent) require computing the null space of the Hessian matrix which is computationally expensive.
To me, it is not entirely clear how much bigger this cost is, and if this is only a problem for large "m". Discussion would be appreciated.


**Q7 Justification For Your Score:**

I am by no means qualified or familiar enough with the area to fully judge this paper.
The paper seems to provide a novel and technically sound approach to solving the proposed problem provides some necessary theory and demonstrates that the approach practically works.


**Q9 Complying With Reviewing Instructions:**

1: Yes.

---

### Official Review · Reviewer_o234 · 2022-04-12

**Q2(1) Originality/Novelty:** 3
**Q2(2) Significance/Impact:** 3
**Q2(3) Correctness/Technical Quality:** 3
**Q2(6) Clarity Of Writing:** 3
**Q6 Overall Score:** 6
**Q8 Confidence In Your Score:** 3

**Q1 Summary And Contributions:**

Related to the optimization problem in a Pareto set, the paper propose an approximation algorithm called Pareto Navigation
Gradient Descent (PNG) with theoretically guaranteed descent and convergence property to solve OPT-in-Pareto.
experiments sho0w its effectiveness.

Overall, the paper is a nice contribution and well written. While the topic is not specific to UAI, it belongs IMHO to operational research, the method may find many applications in problems addressed by the UAI community.


**Q2 Assessment Of The Paper:**

More detailed information regarding each of these aspects is given below:

**Q2(4) Quality Of Experiments (Optional):**

3: Good: The experimental evaluation is adequate, and the results convincingly support the main claims.

**Q2(5) Reproducibility:**

3: Good: Key resources (e.g., proofs, code, data) are available and key details (e.g., proofs, experimental setup) are sufficiently well-described for competent researchers to confidently reproduce the main results.

**Q3 Main Strengths:**

Overall, the paper is a nice contribution and well written. The method may find many applications in problems addressed by the UAI community.


**Q4 Main Weakness:**

The topic is not specific to UAI as it belongs IMHO to operational research.


**Q5 Detailed Comments To The Authors:**

None. The conceptual idea appears clear to me, though I didn't verify it in depth.

**Q7 Justification For Your Score:**

The topic is relevant in the sense of its potential application in multi-objective optimisation problems that can be found here and there within the research problems addressed by the UAI community

**Q9 Complying With Reviewing Instructions:**

1: Yes.

---

### Official Review · Reviewer_9qhX · 2022-04-21

**Q2(1) Originality/Novelty:** 3
**Q2(2) Significance/Impact:** 3
**Q2(3) Correctness/Technical Quality:** 3
**Q2(6) Clarity Of Writing:** 3
**Q6 Overall Score:** 7
**Q8 Confidence In Your Score:** 2

**Q1 Summary And Contributions:**

The authors study the problem of finding a solution for a given multicriteria optimization problem that would, at the same time, minimize a certain value and still be on the Pareto front of the original problem. They also generalize this problem to finding several solutions that all are on the Pareto front, and jointly minimize a certain function (e.g., corresponding to the diversity).

**Q2 Assessment Of The Paper:**

More detailed information regarding each of these aspects is given below:

**Q2(4) Quality Of Experiments (Optional):**

3: Good: The experimental evaluation is adequate, and the results convincingly support the main claims.

**Q2(5) Reproducibility:**

3: Good: Key resources (e.g., proofs, code, data) are available and key details (e.g., proofs, experimental setup) are sufficiently well-described for competent researchers to confidently reproduce the main results.

**Q3 Main Strengths:**

1) The studied topic seems to be quite important---finding a coverage of the Pareto front is an extremely natural goal.

2) The provided empirical results show that the solution provided works

3) The theoretical results are, in a sense, minimal (they show that the method works), but this still seems to be more than a typical situation in the area of the paper.

**Q4 Main Weakness:**

1) There is no comparison of running time in the main body of the paper.

2) I find it difficult to assess to what class of problems the method can really be applied (partially my own problem, since I am not an expert in the area)

**Q5 Detailed Comments To The Authors:**

This is a last minute review by a non-expert, so do not expect too much real insight, but nonetheless I will write what I think.

1. Proofs of theorems:
All the proofs are deferred to an appendix (which, instead of being provided with the paper is in the supplementary part). I would have much appreciated at least one or two of the proofs to be in the main body

2. Related work:
Location of Section 6, Related Work, in the middle of the paper is highly nonstandard. If it cannot be placed early, then I would suggest moving it to the end.

3. Details of Algorithm 1:
On p. 4, right, you write "We include the details of the practical implementation in Algorithm 1" but I honestly could not see much detail there... certainly nothing that would really make it possible for me to implement your algorithm. Some different phrasing might be better.

Some typos:

p. 1, right: "though IT has not been" (??)

p. 2, right: "if there must exists" <-- delete "must"

p. 2, right: "is an gradient-based" --> "is a gradient-based"

p. 3, left: "design F to encourages" --> "design F to encourage"

p. 3, right: "give an uniform" --> "give a uniform"

p. 3, right: "that well cover the" --> "that cover the"

p. 4, left: "rule of form" --> "rule of the form"

p. 4, left, in item iii) you refer to function g( \phi ) first time in a while, so it is nice to point to where it is defined

p. 4, left: "the following optimization PROBLEM:"

p. 7, right: "an small region" --> "a small region"

**Q7 Justification For Your Score:**

The paper seems to be doing good job on an important problem. The math is difficult to follow for me as I am not an expert in the area, but the paper makes a good impression.

**Q9 Complying With Reviewing Instructions:**

1: Yes.

---

### Official Review · Reviewer_tNxS · 2022-04-22

**Q2(1) Originality/Novelty:** 2
**Q2(2) Significance/Impact:** 2
**Q2(3) Correctness/Technical Quality:** 3
**Q2(6) Clarity Of Writing:** 4
**Q6 Overall Score:** 7
**Q8 Confidence In Your Score:** 2

**Q1 Summary And Contributions:**

This paper studies the problem of finding Pareto models that optimize one additional reference criterion, OPT-in-Pareto. (One typical class of such criterion in practice is the diversity measure.) While second-order approaches such as manifold gradient descent are expensive, this paper proposes a first-order approximation algorithm, Pareto navigating gradient descent (PNG). In a way, PNG is an extension of multiple gradient descent (MGD) by Desideri, as it covers MGD as a special case.



**Q2 Assessment Of The Paper:**

More detailed information regarding each of these aspects is given below:

**Q2(4) Quality Of Experiments (Optional):**

3: Good: The experimental evaluation is adequate, and the results convincingly support the main claims.

**Q2(5) Reproducibility:**

3: Good: Key resources (e.g., proofs, code, data) are available and key details (e.g., proofs, experimental setup) are sufficiently well-described for competent researchers to confidently reproduce the main results.

**Q3 Main Strengths:**

- It proposes an approximated solution to a useful and challenging problem.

- Extensive discussions on related work. Good for readers new to the area.

- It is a comprehensive paper, in the sense that it proposes a new algorithm backed by both theoretical and empirical justifications. It also positions the proposed method fittingly among related work.



**Q4 Main Weakness:**

While I don't question the usefulness of the results, but if I understand correctly, the proposal is largely based on the existing MGD approach. I am not sure how novel the new proposal it. Maybe other reviewers can better answer this question.



**Q5 Detailed Comments To The Authors:**

Only minor things:
- Table 1. The best ACC on Fashion-MNIST under IGD+ is achieved by MGD, not PNG.
- The standard deviations reported on ResNet-18 are very small comparing to other results. Any justification?


**Q7 Justification For Your Score:**

I think overall it is a good paper with solid work.

**Q9 Complying With Reviewing Instructions:**

1: Yes.

---

### Decision · Program_Chairs · 2022-05-15

**Decision:**

Accept (Poster)

**Comment:**

Meta Review: The paper proposes a new method to explore the Pareto and finding appropriate new points over this front. From the reviews, the procedure appears novel (to the extent reviewers can judge) and technically solid.

The main critic done to the paper, that also transpire in the reviewer comments is that this paper is marginally in the scope of UAI, as its main topic concerns multi-objective optimisation (admittedly applied to learning, yet learning and uncertainty reasoning are not necessarily intertwined), with only a limited connection to uncertainty handling (in contrast to other papers such as, e.g., "Learning Pareto-Efficient Decisions with Confidence").

In summary, this is a nice and original paper, but the UAI audience is not the ideal target for it.